# Fast and versatile sequence-independent protein docking for nanomaterials design using RPXDock

**William Sheffler[1]**, **Erin C. Yang[1,2,3]**, **Quinton Dowling[1,4]**, **Yang Hsia[1]**, **Chelsea N. Fries[1]**, **Jenna Stanislaw[1,2,5,6]**, **Mark D. Langowski[1,7]**, **Marisa Brandys[1,2]**, **Zhe Li[1]**, **Rebecca Skotheim[1]**, **Andrew J. Borst[1,2]**, **Alena Khmelinskaia[1,2,5,6]**, **Neil P. King[1,2]**, **David Baker[1,2,8]** *

1 Institute for Protein Design, University of Washington, Seattle, Washington, United States of America, 2 Department of Biochemistry, University of Washington, Seattle, Washington, United States of America, 3 Graduate Program in Biological Physics, Structure & Design, University of Washington, Seattle, Washington, United States of America, 4 Department of Bioengineering, University of Washington, Seattle, Washington, United States of America, 5 Transdisciplinary Research Area "Building Blocks of Matter and Fundamental Interactions (TRA Matter)", University of Bonn, Bonn, Germany, 6 Life and Medical Sciences Institute, University of Bonn, Bonn, Germany, 7 Graduate Program in Molecular and Cellular Biology, University of Washington, Seattle, Washington, United States of America, 8 Howard Hughes Medical Institute, University of Washington, Seattle, Washington, United States of America

☮ These authors contributed equally to this work.
* dabaker@uw.edu

☮ OPEN ACCESS

**Data Availability Statement:** All data are available in the main text or as supplementary materials. Scripts, computational methods, and design models are available on GitHub at https://github.com/willsheffler/rpxdock. For O43-rpxdock-EK1,

## Abstract

Computationally designed multi-subunit assemblies have shown considerable promise for a variety of applications, including a new generation of potent vaccines. One of the major routes to such materials is rigid body sequence-independent docking of cyclic oligomers into architectures with point group or lattice symmetries. Current methods for docking and designing such assemblies are tailored to specific classes of symmetry and are difficult to modify for novel applications. Here we describe RPXDock, a fast, flexible, and modular software package for sequence-independent rigid-body protein docking across a wide range of symmetric architectures that is easily customizable for further development. RPXDock uses an efficient hierarchical search and a residue-pair transform (*RPX*) scoring method to rapidly search through multidimensional docking space. We describe the structure of the software, provide practical guidelines for its use, and describe the available functionalities including a variety of score functions and filtering tools that can be used to guide and refine docking results towards desired configurations.

## Introduction

There has been considerable progress in the design of symmetric protein assemblies ranging from relatively small, cyclically symmetric proteins, to megadalton structures containing more than 100 subunits [1–11]. There are three widely used approaches for generating such materials: generation of backbone arrangements using parametric equations (primarily applied to

coordinates are deposited in the Protein Data Bank with the accession code 8FWD; the cryo-EM density map is deposited in the Electron Microscopy Data Bank (EMDB) with the accession code EMD-29502.

**Funding:** Funding for this work was provided by the Audacious Project at the Institute for Protein Design (N.P.K. and D.B.), The Open Philanthropy Project for Improving Protein Design Fund (D.B.), NSF DGE-1762114 (E.C.Y.), the Bill & Melinda Gates Foundation grant #INV-010680 (N.P.K. and D.B.), a Rosetta Commons Post-Baccalaureate Fellowship (J.S.), National Science Foundation grant CHE-1629214 (N.P.K. and D.B.), grant DE-SC0018940 MOD03 from the U.S. Department of Energy Office of Science (A.J.B., D.B.), the National Institutes of Health grants 1P01AI167966 (N.P.K.) and P50AI150464 (N.P.K.), and the Howard Hughes Medical Institute (D.B.). The funders had no role in study design, data collection and analysis, decision to publish, or preparation of the manuscript.

**Competing interests:** I have read the journal's policy and the authors of this manuscript have the following competing interests: "A provisional patent application has been filed (63/459,822) by the University of Washington, listing E.C.Y., Y.H., W.S., Z.L., Tim Huddy, N.P.K., and D.B. as inventors.

helical bundles with cyclic symmetries such as coiled coils) [12–14]; rigid fusion of cyclic protein oligomers with their internal symmetry axes aligned with those of a desired symmetric architecture [6–8,15,16], and sequence-independent rigid body docking of cyclic oligomers such that their internal symmetry axes are aligned with those of a desired architecture followed by combinatorial sequence optimization at the newly generated protein-protein interface to drive assembly [3,4,17–22]. The third approach has the advantage of considerable generality since cyclic building blocks can be combined in a very wide variety of docked arrangements independent of the constraint of chain fusion accessibility. However, while many sequence-dependent docking methods exist for protein-protein interaction prediction [23–27], software for sequence-independent docking for protein design remains relatively underdeveloped. One challenge such methods face is that in the absence of sequence information, scoring of different docked arrangements is not straightforward. Fast Fourier Transform (FFT) docking methods can be used without sequence information for design applications, but the interatomic interactions are blurred out, and the results are generally not rotationally invariant [28]. The "slide-into-contact" `tc_dock` method [19] and derivatives thereof, which use a residue-pair transform (*RPX*) hashing method to approximate residue-residue interaction energies prior to explicit sequence design [17], have proven useful in the design of a wide variety of symmetric protein nanomaterials including cyclic homooligomers [17], dihedral assemblies [18], multi-component symmetric protein nanocages [1–4,19], one-dimensional fibers [20], two-dimensional layers [21,22], and three-dimensional crystals [29]. However, these methods have not been thoroughly documented, are computationally inefficient, and are difficult to modify for new applications.

We set out to develop a computationally efficient and readily customizable method for rigid body sequence-independent docking capable of pruning unproductive regions of the available search space to reduce time spent in computationally expensive downstream sequence design calculations. Here we describe the RPXDock software, which improves on the earlier `tc_dock` software in three major areas:

1. *Generalizability*: RPXDock unifies previous docking methods specific to particular architectures under a single framework that globally searches rigid body space, sampling the relevant rigid body degrees of freedom (DOFs) across multiple classes of symmetric and asymmetric architectures.

2. *Extensibility*: All the computationally expensive operations in RPXDock are written in C++ that the user interfaces via python. The lower-level libraries are interoperable and thoroughly covered by tests. The codebase is structured to encourage development of new user-defined constraints such that the top outputs are the highest quality docks that satisfy a given set of criteria. For example, newly implemented features allow biasing of the results towards particular interface sizes and protein termini geometry. Adding new docking architectures, score functions, or filters requires minimal updates to existing code.

3. *Speed*: RPXDock utilizes hierarchical decomposition of the underlying degrees of freedom paired with a matching hierarchy of *RPX* score functions to rapidly scan a full docking space at lower resolution; discard large, low-quality regions of the space; and refine docks in progressively higher-quality regions. As a result, RPXDock is very fast and computationally inexpensive, capable of explicitly evaluating millions of docked configurations in minutes. A typical docking trajectory involving two building blocks finishes in seconds to minutes, including overhead.

Prior to publication of this manuscript, RPXDock was used to successfully design cyclic oligomers [30], one-component nanocages [31], two-component nanocages [29], and even larger

pseudo-symmetric nanomaterials, establishing its utility and generality. Here we provide a guide to using RPXDock to produce rigid body docks, prior to sequence design [32,33]. Additional technical descriptions of individual modules in the software are provided in the S1 Text.

## Design and implementation

### Overview of RPXDock general methodology

A visual outline of the software structure is provided in **Fig 1**. Users pass options into the `dock.py` application, which include required inputs such as Protein Data Bank (.pdb) files and the desired docking architecture, as well as other optional docking parameters described in detail in subsequent sections. A full list of command line options can be found in **S1 Table** and can be retrieved interactively using `--help`. The `dock.py` application interprets user-defined options and drives the machinery behind the docking algorithm. Input .pdb files are loaded using PyRosetta [34] as poses, then converted by the *Body* class into body objects. Various structural data are compiled from the input .pdb files, including transformable Bounding Volume Hierarchies (BVH) that index atomic coordinates. The *Spec* and *Sampler* classes define the DOFs of the target architecture and how they are to be broken down hierarchically. This space is traversed in the *Search* class, using a hierarchical search algorithm similar to branch-and-bound search [35]. During each iteration of the hierarchical search, each docked configuration, or transform, is evaluated by a residue-pair motif score [17] matched to the resolution of the search step, and then by a user-selected score function. Residue-pair motifs are identified by interacting pairs of backbone positions determined via the BVH data structures. Once the hierarchical search algorithm reaches its final resolution, the remaining docked configurations can be filtered with optional user-defined metrics. The filtered docked configurations are clustered based on redundancy among docked transforms and stored by the *Result* class as transformation matrices and scores in an xarray dataframe. The *Result* class can subsequently be used to re-apply a transformation matrix to the stored input pose, yielding a full-atom.pdb file.

### Inputs and bodies

RPXDock uses the PyRosetta [34] *pose* module to load the atomic coordinates of input .pdb files and make secondary structure assignments via Define Secondary Structure of Proteins (DSSP) [36,37]. The PyRosetta pose is stored in the *Body* class as a Body object. Input .pdb files are provided to the `dock.py` application using the `--inputs1` option. The input can be a path to a single .pdb (e.g., example.pdb) file, or a path with a wildcard (e.g., /path/to/files/*.pdb) can be supplied for multiple inputs. For multicomponent docking, additional inputs can be provided using the `--inputs2` and `--inputs3` option as necessary. For trajectories with multiple input lists provided to `--inputs[n]`, each object in the list will be sampled against every other object in a partner list. The results for list inputs are batched and ranked together against one another. Thus, the "top" results may not include representatives from every input .pdb. If results from every input are desired, the user can either analyze the entire output list or execute each input or pair of inputs in separate RPXDock trajectories.

### Bounding volume hierarchy (BVH)

The *Body* class implements a Bounding Volume Hierarchy (BVH) representation for efficient contact, sliding, clash checking, and determination of contacts for scoring [38]. As time taken for these operations scales with interface size, valuable compute time is saved by our implementation of BVH, which utilizes spheres rather than traditional bounding boxes for

# dock.py Application

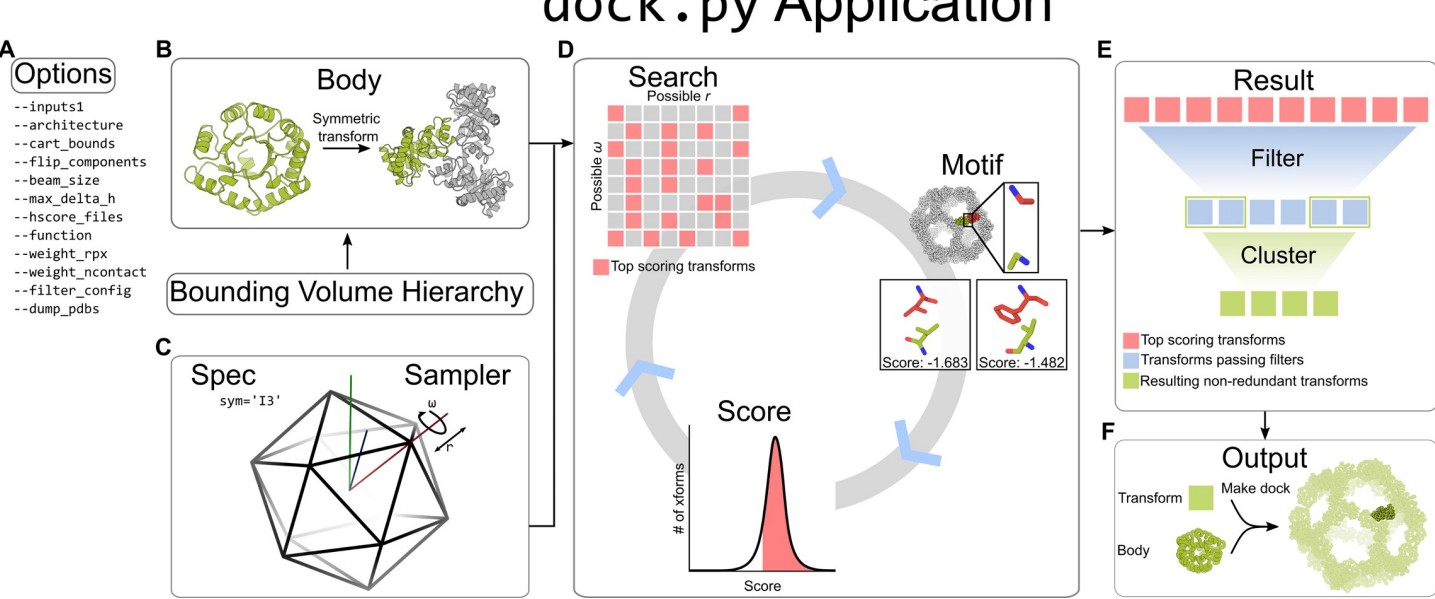

**Fig 1. General software structure of RPXDock. A.** User-defined inputs are given as options to the dock.py application. **B.** Within the application, input .pdb files are stored in the Body object as a PyRosetta pose. The *Body* class implements a Bounding Volume Hierarchy (BVH) for rapid operations on coordinates. **C.** The *Spec* and *Sampler* classes define the rigid-body DOFs the Body object is allowed to sample. **D.** Within the *Search* class, the Body object receives the DOFs as rigid body transforms (indicated as grid squares). Each transform is evaluated by the *Motif* and *Score* classes, which ranks the quality of residue-pair motifs at a given interface of a dock [17] and subsequently summarizes the residue-pair motif scores with additional interface quality metrics through a user-selected score function. The top scoring transforms are searched iteratively with higher resolution sampling and scoring in a hierarchical search algorithm. **E.** The final top scoring transforms from the search are fed into the *Result* class, which prunes the results using filter metrics and clusters the transforms based on backbone redundancy. **F.** The results are stored and output as transforms, which can be re-applied to the input Body object to generate a full-atom .pdb file of the resulting docked configuration.

rotational invariance, allowing rigid body motions without recalculation. The BVH is first used to check for contacts, rapidly discarding configurations where bodies do not interact. During docking, clashing docks where the BVH intersect are removed. Lastly, BVH identifies all interacting pairs of residue stubs during scoring so that only interacting residues are evaluated. These operations are adjusted conservatively based on the resolution of the sampling, such that even large regions of the search space can be discarded as unlikely to contain favorable configurations.

## Defining degrees of freedom (DOFs)

Sampling configurations of bodies is performed through a composable set of primitive samplers, including 1D, 2D, and 3D cartesian grids, 1D rotations, 2D directions, and 3D orientations. The space of orientations is modeled as the equivalent space of quaternions on a 3-sphere, and sampling is performed by subdividing the cells of a bitruncated 24-cell, a uniform 4D polytope that divides the 3-sphere uniformly into roughly cubic regions. This approach avoids the pitfalls of using Euler angles to represent 3D rotations. Streamlined combinations of these samplers are provided, such as rotation and translation on a symmetry axis, or a full 6D rigid body transformation, as well as a simple framework to create user-defined compositions and products of sampling spaces. All of these samplers and their combinations provide configurable resolutions, bounds, a hierarchy of nested sampling grids, and the ability to map indices between higher and lower resolutions.

## Symmetric architectures

In symmetrical systems, the "architecture" defines the connectivity and allowed rigid-body kinematics, or movements, of the building blocks. RPXDock currently has built-in support for asymmetric, cyclic, stacking, dihedral, wallpaper (2D), and polyhedral group architectures. While the current release of dock.py does not support helical (1D) and crystal (3D) architectures, the components necessary for these protocols are available, and we plan to implement these in future builds of RPXDock. The desired architecture is specified per trajectory with the --architecture option using a keyword (**Table 1**).

## Input preparation

To dock two distinct monomers asymmetrically or to form cyclic oligomers, monomeric building blocks should have their center of mass at [0,0,0] (**Fig 2A and 2B**). RPXDock will not center the inputs by default, but the --recenter_input option can be passed to translate a monomeric building block such that its center of mass is at [0,0,0]. The final transform values reported are relative to the recentered pose, so it is recommended that inputs are pre-centered if the user plans to use these values.

To form dihedral, stacking, wallpaper, and polyhedral group symmetries such as tetrahedral, octahedral, and icosahedral architectures, the input building blocks must be cyclic oligomers. The input .pdb files must be pre-aligned such that their internal rotational symmetry axes are aligned to the Z axis and the center of mass of the oligomer should be centered at [0,0,0] (**Fig 2C and 2D**). It is important to note that the input .pdb files should only contain the asymmetric unit (asu) of the cyclic oligomer rather than the full symmetric building block, as RPXDock will generate the symmetry-related chains. Currently, dihedral docking only supports one-component (i.e., homomeric) architectures; stacking supports two-component architectures; polyhedral group docking supports one-, two-, and three-component architectures; and wallpaper docking supports two- and three-component architectures.

**Table 1. Keywords associated with each currently supported architecture.**

| --architecture | Number of unique protein components supported | |
|---|---|---|
| **Asymmetric** | 2 | "ASYM" |
| **Cyclic** | 1 | "C[n]" where [n] = 1, 2, 3, ..., n |
| **Stacking** | 2 | "AXLE_[n]" where [n] = 1, 2, 3, ..., n, or "AXLE_1_[m]_[n]" where [m] and [n] correspond to the cyclic symmetries of the inputs and [n] ! = [m]. Currently supports up to [m] = 5 and [n] = 6. |
| **Dihedral** | 1 | "DX_X", where X is the cyclic symmetry perpendicular to the dihedral plane and the oligomeric state of the input scaffold |
| | 1 | "DX_2", same as above, but the input oligomer is a dimer aligned to the dihedral plane |
| **Polyhedral group** | 1 | "T2", "T3", "O2", "O3", "O4", "I2", "I3", "I5" |
| | 2 | "T32", "T33", "O32", "O42", "O43", "I32", "I52", "I53" |
| | 3 | "T332", "O432", "I532" |
| **Wallpaper** | 2, 3 | "P6_632", "P6_63", "P6_62", "P6_32", "P3_33", "P4_42", "P4_44" where "Px" describes the lattice symmetry and cyclic oligomer symmetries are listed after the underscore |

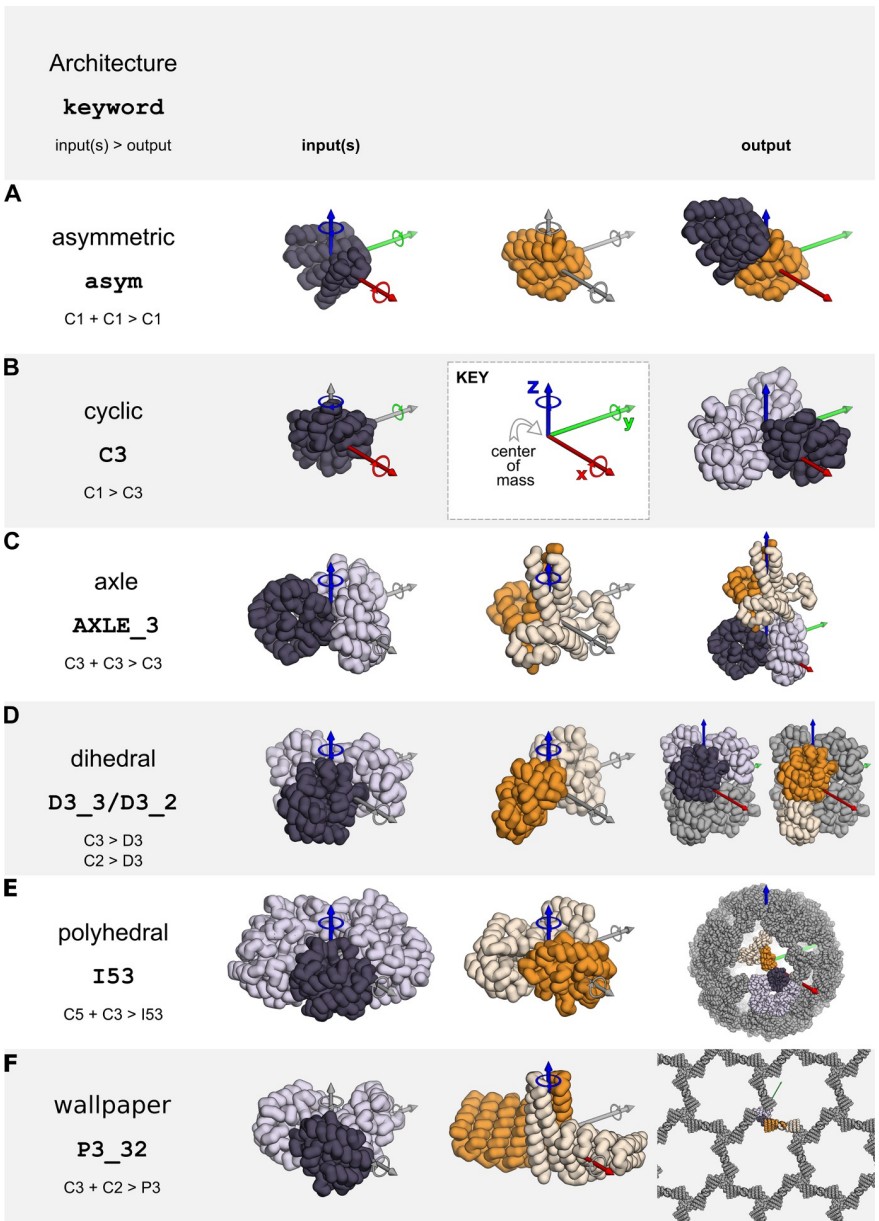

**Fig 2. Example inputs and docking output architectures currently supported by RPXDock.** X/Y/Z cartesian axes are shown in red, green, and blue respectively. Corresponding translational and rotational DOFs are sampled along and around these axes. Axes where DOFs are not sampled for an architecture are colored gray. **A.** Asymmetric docking samples 6 DOFs belonging to the first of two input monomers. **B.** Cyclic docking samples four DOFs belonging to an input monomer to generate a cyclic structure with its cyclic axis aligned to the Z axis. **C-F.** Oligomeric input structures must have their cyclic axis aligned to the Z axis and the input .pdb should only contain the asu (dark). Stacking, dihedral, polyhedral group, and wallpaper docking samples the rotational and translational DOF along the Z axis of the input cyclic oligomer, which is aligned during docking to the relevant rotational symmetry axes in the target architecture.

## Defining the search space

The search spaces for the supported architectures in RPXDock are either one-, two-, or three-body problems and the number of allowed DOFs sampled depends on the kinematics defined by the specified architecture. Two-body asymmetric docking technically allows all three

rotational DOFs and all three translational DOFs (X, Y, and Z) per component, but in practice it is sufficient to hold one component static while sampling the other component against it (**Fig 2A**). Cyclic docking allows sampling of all three rotational DOFs but only one translational DOF (the radius), as sampling the remaining two cartesian DOFs results in identical final structures (**Fig 2B**). Each oligomeric component in stacking, dihedral, polyhedral group, wallpaper, and crystal architectures is aligned to a single rotational symmetry axis in the target architecture (the Z axis in the input .pdb) and is therefore limited to sampling one rotational and one translational DOF along that axis (**Fig 2C–2F**).

Each translational or rotational DOF is set by bounds in cartesian or angular space. Cartesian bounds can be set by `--cart_bounds d1 d2` where the lower (`d1`) and upper (`d2`) bounds are distances in Ångstroms. The default values of `d1` and `d2` for symmetrical architectures are `0` and `500`, limiting the search to only the positive direction of the space, as the reverse translational degrees of freedom are redundant when combined with the `--flip_components` option (see below). For asymmetrical docking scenarios, however, the default values are `-500` and `500`, allowing search in both directions. The larger this range is set, the longer the runtime and memory required. Thus, if the user has an idea of the desired search size, these values should be reduced as appropriate. Angular bounds are defined by the cyclic symmetry of the input component by default. For example, the angular bounds of a C3 input component are `0` and `120°`. The final search space is defined by combining the DOF assignments and boundaries.

## Restricting additional DOFs

For some docking problems, a user may want to restrict either one or all of the rotational or translational DOFs of their inputs during the search; for example, some docking problems require specific building blocks to be aligned to additional symmetry axes [29]. The rotational and/or translational DOFs can be turned off (`--fixed_rot`, `--fixed_trans`, `--fixed_components`) or restricted to a user-defined range (`--fixed_wiggle`). These are activated by listing which inputs should be fixed (0-delimited; e.g., `--fixed_rot 1` to restrict the rotation DOF of all .pdb files provided in `--inputs2`, or `--fixed_rot 0 1` to restrict the rotation DOF of all .pdb files provided in `--inputs1` and `--inputs2`).

- `--fixed_rot`: fix the rotational DOF for desired input component

- `--fixed_trans`: fix the translational DOF for desired input component

- `--fixed_components`: fix both the translational and rotational DOFs for desired input component

- `--fixed_wiggle`: limit the translation and rotational DOFs to a certain range from the starting position. Additionally, specifications for the upper and lower bounds (ub and lb) of translation and rotation are required (`--fw_rot_lb`, `--fw_rot_ub`, `--fw_trans_lb`, `--fw_trans_ub`), where the upper and lower bounds are not equal.

The `--flip_components` option can be used to specify which cyclic components proceed to DOF sampling both before and after rotating the input .pdb 180° along the X axis ("flipping"). For example, a C3 oligomer can sample along the Z axis 0 to 120° and also 0 to 120° after flipping. This is effectively identical to sampling "negative" translations in dihedral, polyhedral group, and stacking architectures, and is required to fully search the available docking space in most symmetries. This option takes a list of boolean values corresponding to each input and defaults to *true* for all components (e.g., `--flip_components 1 1` for a trajectory with two inputs).

## Sampling the search space

RPXDock samples the defined search space via the modular sampler objects previously discussed and stores transformation matrices for each component of a set of sampled docked configurations. Each transform is applied to the respective input(s), resulting in a single docked configuration that was sampled during a docking trajectory, and is subsequently used to check for clashes and in some cases "flatness" at each iteration of the search. Clashing is evaluated by the BVH as described above. The "flatness" of a docked configuration is calculated during cyclic and multi-component docking (i.e., polyhedral group, stacking, wallpaper). During cyclic docking, flatness refers to the orientation of the longest physical axis of the input .pdb, as defined by principal component analysis, relative to the cyclic symmetry axis. The "flatness" of a cyclic dock can be constrained using the `--max_longaxis_dot_z` option, which restricts the orientation of the input .pdb relative to the cyclic symmetry axis (conventionally aligned along Z) by calculating the cosine between this axis and the longest axis of the input .pdb. Docks that exceed the cosine value given by the `--max_longaxis_dot_z` option are removed from the next stage of the search. This option can be set to any value between 0 and 1 (inclusive), where 1 allows the input .pdb to adopt any configuration relative to the cyclic symmetry axis, while 0 constrains the long axis of the input .pdb to perfect alignment, or perpendicularity, to the cyclic symmetry axis. During multi-component docking, flatness refers to differences in the translation of each component along its respective symmetry axis. In this case, the `--max_delta_h` option can be used to set an upper bound on the maximum allowable difference in offset between components.

**Global and hierarchical search.** In the asymmetric two-body docking problem, there are six DOFs: three translational and three rotational, where one body samples all six DOFs while the other remains static. As all six DOFs are sampled explicitly, the total number of transforms to evaluate equals the number of top-level samples (which is determined by the type and resolution of each DOF) multiplied by the number of subdivisions of that DOF raised to the 6th power. For example, if a typical top-level search space with six dimensions comprised of 10,000,000 total samples is used, sampling a single transform in a 16 Å space at a resolution of 16 Å for each dimension would result in $10{,}000{,}000 * 1^6 = 10{,}000{,}000$ total transforms across the entire search space. Sampling at this 16 Å space at 1 Å resolution for each dimension (16 transforms per dimension) would result in $10{,}000{,}000 * 16^6 = 167$ trillion total transforms to sample the entire search space. Enumerative sampling, even with some dimensionality reduction as implemented in previous iterations of "slide into contact" docking, is prohibitive at a reasonable resolution in architectures with a high number of DOFs [17,19,39,40].

To enable efficient exploration of the search space in such architectures, we implemented an iterative hierarchical search that prunes away areas of the search space unlikely to contain good solutions [41,42]. In this sampling and evaluation scheme, the search begins at low resolution and is repeated with increasing resolution at each iteration. Only the top-scoring regions of the search space are kept for further exploration in the next iteration (**Fig 3A and 3B**). This reduces the number of samples that must be evaluated at each stage such that the total number of transforms evaluated no longer grows exponentially with dimension. For the simple 2D illustration in **Fig 3A**, the total number of samples is reduced from 256 to 24. Efficiency gains are roughly exponential with dimensionality and are thus much higher for less constrained docking problems. At each resolution, configurations are scored as an implicit ensemble (**Fig 3B**) through the use of residue-pair motifs (see "Score functions and motifs" section), tuned to provide an approximation of the best possible score within a corresponding ensemble of residue pair positions (**Fig 3C**). By evaluating the best possible score within an

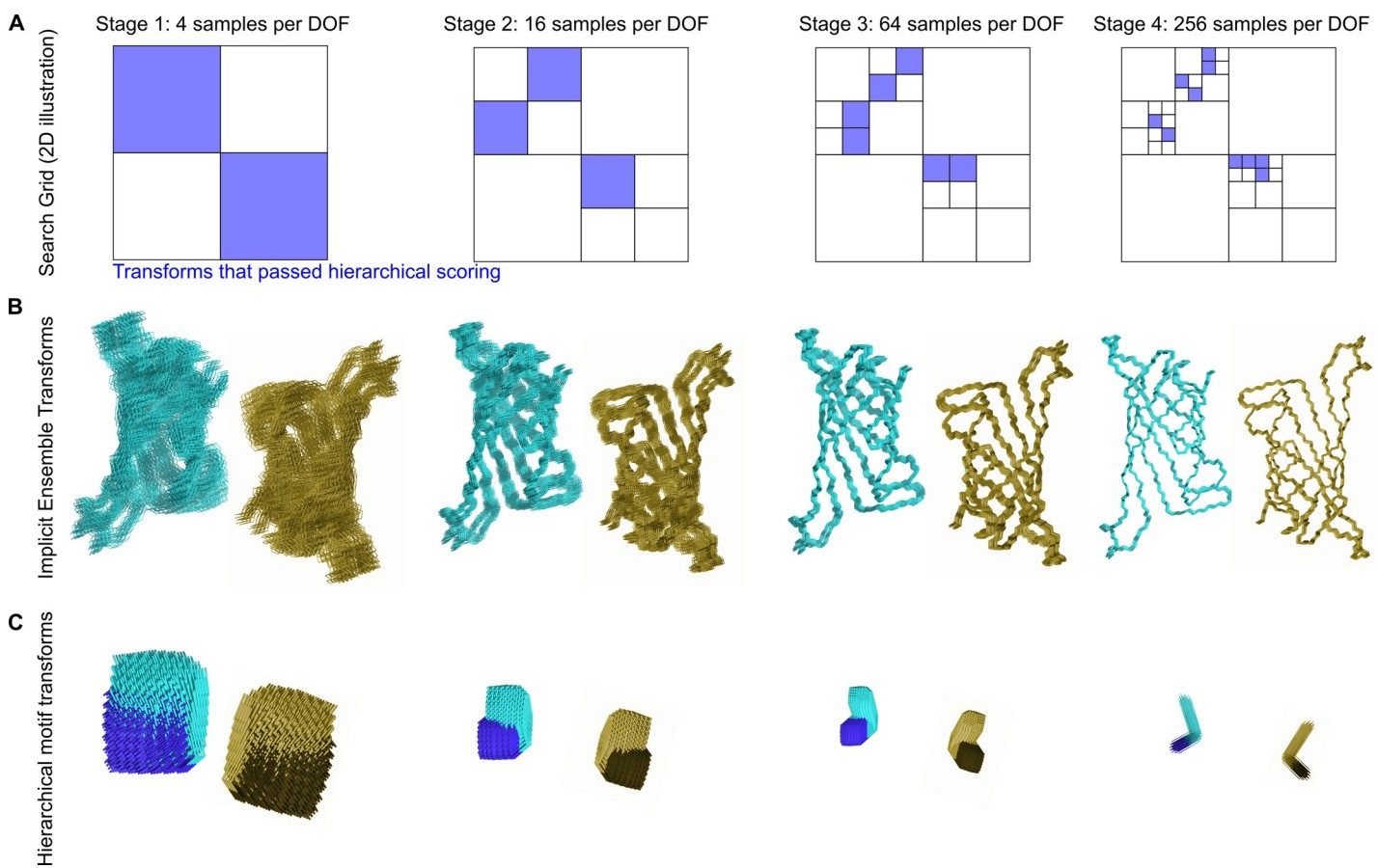

**Fig 3. Schematic representation of hierarchical sampling. A**. Schematic of a search grid for a single DOF keeping only the transforms that passed hierarchical scoring (blue) at each stage of search. This reduces the space searched at later stages where the search grid is subdivided at increasing resolution. **B**. A schematic depiction of protein backbones sampled with increasing resolution. The backbones shown would correspond to a single blue box at each stage of the search depicted in panel A; a cloud of such backbones would be sampled for each of the distinct docked configurations corresponding to each blue box. **C**. Residue-pair motifs are also evaluated at increasing resolution during each iteration of the search.

ensemble, as opposed to an average score, an entire region of docking space can be discarded without missing desirable docks.

Due to the reduction in search space, the hierarchical search and scoring of a typical system with the default search space parameters takes approximately 30 seconds on a 4-core cpu. Further reductions or expansions in the number of transforms sampled at each stage of the hierarchical search protocol can be implemented using the `--beam_size` option, which defines the maximum number of sampled docks taken to the next stage of a hierarchical search protocol (default 100,000). The `beam_size` excludes the first, most coarse stage, which samples the entire search space at the lowest assigned resolution, as defined by `--ori_resl` (default 30˚) and `--cart_resl` (default 10 Å).

To reduce low-resolution artifacts, we take the upper bound of scores within each grid square in the hierarchical search rather than its potentially low scoring and non-representative low resolution center (average). This effectively gives each grid square the "benefit of the doubt" during iteration, so that poor scoring regions can be discarded with confidence. We found during empirical testing that the hierarchical search approach did not over-prune substantial numbers of "good" candidates (**S1 Fig**). Specifically, we compared how efficiently the hierarchical sampling method recovered the top docks identified by enumerative grid

sampling (~$10^8$ total asymmetric docks) (**S1A Fig**). The hierarchical search method recovered the top dock in this test set by searching less than $1/10^5$ of the total search space and the top 10 docks by searching less than $1/10^2$ (**S1B Fig**). To find the top 100 and top 1000 docks, the hierarchical method searched nearly the entire space, although it recovered 80% of the top 100 docks within $1/10^4$ of the total search space. As identifying ~10 top docks per input or input pair is reasonable for most docking problems in practice, the reduction in search space and the consequent reduction in time that the hierarchical search method requires to find top-scoring docks is most likely an acceptable tradeoff.

While a major advantage of RPXDock is utilizing the hierarchical search method (`--docking_method hier`), it is possible to globally search the conformation space (`--docking_method grid`). This option may be appropriate for one-component dihedral or polyhedral docking problems that have two or fewer DOFs available. As the global search space is sampled at a single resolution, the user should specify different search resolutions in translational and rotational DOFs (`--grid_resolution_cart_angstroms` and `--grid_resolution_ori_degrees`) across multiple independent trajectories. Nevertheless, grid search is implemented mainly for debugging purposes and is not recommended for production runs.

**Specifying termini direction and accessibility.**    For polyhedral group architectures, the orientation and accessibility of the components' termini can be important for downstream applications such as multivalent antigen display via genetic fusion [1,39,40,43,44]. Two options are implemented for polyhedral group architectures to (1) restrict the sampling space to docks with termini in the desired orientation (`--termini_dir[n]`) and (2) evaluate the accessibility of the termini residues (`--term_access[n]`). The `--termini_dir[n]` and `--term_access[n]` options mirror the syntax of the `--inputs[n]` options, where `--termini_dir1` and `--term_access1` refer to the termini direction and accessibility for `--inputs1`, `--termini_dir2` and `--term_access2` for `--inputs2`, etc. Both options operate by aligning an ideal 21-residue alpha helix to 7 residues at the user-specified termini.

The `--termini_dir[n]` option evaluates the helix orientation by calculating the Z direction of the vector defined by the center of mass of the first and last three residues of the aligned ideal helix (e.g. residues 1–3 to 19–21 for N termini, and inversely for C termini). The option then picks the desired orientation from the required `--flip_components` option and disables sampling of the other. The aligned ideal helix is removed before sampling docked transforms.

The `--term_access[n]` option evaluates the accessibility of user-defined termini during sampling by adding the aligned ideal helix to the *Body* class for the BVH to use for clash checking at each step of the search. The aligned helix is omitted in the *Score* and *Result* classes for *RPX* scoring and output. The option syntax is as follows:

- `--termini_dir1 [--termini_dir2,--termini_dir3]`: Accepts a desired orientation as "in", "out", or "None", for the amino terminus followed by the carboxyl terminus (space-delimited) for each corresponding input (`--inputs1` for `--termini_dir1`, `--inputs2` for `--termini_dir2`, etc.). "In" restricts sampling to configurations in which the specified terminus points towards the architecture's center of mass, while "out" restricts sampling to the opposite. This option alternatively accepts a space-delimited pair of booleans where "in" is *True*, and "out" is *False*. The option(s) default to "None". The `--flip_components` option must be set to *True*.

- `--term_access1 [--term_access2,--term_access3]`: Accepts a space-delimited pair of boolean values to enable evaluation of terminus accessibility at the amino

and carboxyl termini of each component, respectively. (e.g. `--term_access1 0 1` evaluates accessibility of the carboxyl termini for input .pdb files passed through `--inputs1`)

### Evaluating docked configurations

**Residue-Pair Transform (*RPX*) Scoring.**   We employ a 6D implicit side-chain methodology when evaluating residue-pair interactions in a sequence-independent manner. The interaction between two residues is represented by the full 6D rigid-body transformation between their respective backbone N, Cα, and C atoms [17]. Transforms are binned into six dimensional body-centered cubic lattices, with three dimensions each for translation and rotation. The curved space of rotations is divided into 24 relatively flat cells, with one lattice in each cell. A pre-compiled residue-pair transform, or *hscore*, database of all residue-pair interactions for each amino acid found in structures from the Protein Data Bank (PDB) was binned based on this method and scored using the Rosetta full-atom energy function [45]. During docking, pairs of residues across a docked interface are assigned an *RPX score*, which is the lowest precalculated Rosetta full-atom energy found in the relevant spatial transformation bin of the database. The top-scoring residue pair scores across the interface are evaluated based on a user-defined RPXDock score function (see Ranking dock quality (score functions)). This score was previously found to be more predictive of the interface energy from full-atom sequence-design calculation than the Rosetta centroid energy function or other "coarse-grained" scoring models [17].

**Motif-enriched docking.**   A user may want to diversify or restrict the motifs and secondary structure elements used to score RPXDocked configurations. This can be done using the `--hscore_files` and `--hscore_data_dir` options. The path suffix in `--hscore_files` will be appended to `--hscore_data_dir`, which is the default path to search for *hscore* files. These *hscore* files can be read in as a tarball zipped .txz format that is slow to load but Python version-agnostic, or in a .pickle format that is fast but Python version-dependent. The `--generate_hscore_pickle_files` option can be passed to generate .pickle versions from the .txz file, which can then simply be moved to the corresponding hscore folder before use. Each category of *hscore* files contains scores for a subset of the full residue-pair motif database, restricted to certain amino acid identities and secondary structure elements. By restricting the database, transforms with no motifs found among the chosen subset result in a score of zero, and are thrown out when proceeding to the next phase of the search, thus biasing against the unselected amino acids and secondary structures (H α-helices, E β-sheets, and L loops). Note that RPXDock is sequence-agnostic, meaning the residue identities of the input .pdb are ignored when placing motif pairs. The default motif set only includes pairs involving isoleucine, leucine, and valine; and only in α-helices. The following *hscore* files are pre-compiled and provided in the Institute for Protein Design public repository at https://files.ipd.uw.edu/pub/rpxdock/hscores.zip:

- ILV_H (default; isoleucine, leucine, valine; helices only)

- AILV_H (alanine, isoleucine, leucine, valine; helices only)

- AFILMV_EHL (all hydrophobic amino acids; all secondary structures: sheets, helices, and loops)

**Restricting regions for scoring.**   *Score only SS*. Scoring can be restricted to only certain secondary structure elements using the `--score_only_ss` option (any non-delimited

combination of 'EHL' for sheets, helices, and loops). When active, only residue pairs where at least one of the two motif pairs reside on the desired secondary structure types will be scored. To additionally restrict such that both motif pairs must reside on the designated secondary structure types, the `--score_only_sspair` option can be used. Conceptually this results in a similar effect as providing *hscore* files for only the desired secondary structure types and will enrich for these motifs. Users should note that these restrictions do not explicitly remove or penalize contacts, which contribute to the docking score independently of motifs, at positions on non-desired secondary structure elements.

*Masking (Allowed residues)*. To bias the search towards generating interfaces focused on a specific region of the input structure(s), a list of residue positions can be provided using the `--allowed_residues[n]` option. Specifying positions in this way does not prevent other regions of the protein from forming contacts, nor does it affect clash checking. Instead, regions of the protein structure not included as allowed residues simply do not contribute to the score of the docked configuration, thus biasing the search. The `--allowed_residues[n]` option mirrors the syntax of the `--inputs[n]` options, where `--allowed_residues1` refers to the list of allowed residues for `--inputs1`, `--allowed_residues2` for `--inputs2`, and so forth. The `--allowed_residues[n]` options can either be left blank (default), take a single file which applies to all corresponding component inputs, or take a list of files which must have the same length as the list of inputs. The files themselves must contain a whitespace- and/or newline-delimited list of either numbers and/or ranges using Python syntax. For example, a three-lined file:

1 2 3 4 5

7:12

80:-1

will result in specifying residues 1 2 3 4 5 (first line), 7 8 9 10 11 12 (second line), and 80 through the last residue (third line) as "allowed" for all of the corresponding list of inputs. Residue numbering starts from one and numeric gaps in the input .pdb files are ignored and renumbered sequentially. Multi-chain inputs will be concatenated into a single chain by default. It is recommended that users sanitize input .pdb files to these standards prior to using RPXDock to prevent unexpected results.

**Ranking dock quality (score functions).** RPXDock evaluates dock quality with a score function that summarizes the number of contacting residue-pairs at an interface ("contacts") and the *RPX* score, derived from motif pairs as described above. The *RPX* score is evaluated for each pair of residues in the docked interface within a maximum distance of each other, as defined by the `--max_pair_dist` option (default 8.0 Å at the highest search resolution), which scales with the resolution during the hierarchical search. Afterwards, all the relevant *RPX* scores are combined according to the score function definition, controlled by the `--function` option. The default score function (`stnd`) is defined as:

$$score = a*RPX + b*ncontact$$

where *a* and *b* are coefficients set by `--weight_rpx` and `--weight_ncontact` (default 1 and 0.001, respectively), *RPX* is the sum of the maximum *RPX* scores for each pair of contacting residues (*i*) in the interface ($\sum_{i=1}^{n} max(motif\_score_e)$), and *ncontact* is the number of pairwise contacts in the interface. In this standard score function, *RPX* is highly covariate with *ncontact*, and thus it is also highly correlated with the total score. As a result, because RPXDock

seeks to maximize the score, the standard algorithm will tend to find the largest possible interfaces.

*SASA weighted (sasa_priority) score function.* It is likely that there is an optimum interface size for each docking architecture and the subtypes within them, due to the apparent relationship between interface size and interface strength of symmetrical assemblies, the latter of which can be a critical determinant of the fidelity of the assembly process [46,47]. Therefore, the user may wish to bias docked conformations toward a particular interface size. This can be achieved by taking advantage of the correlation between *ncontact* and interface size, as measured by buried solvent accessible surface area (SASA) [48] (**S2 Fig**). The `sasa_priority` score function seeks to find the best docked configuration for a target interface size as measured by the average motif quality $\underline{X}_{RPX}$ across all residue-pair combinations. For each residue pair, the maximum motif score is considered in this average. Thus, the `sasa_priority` score function is defined as:

$$score = a*_{X_{RPX}} + b*lnN(\mu, \sigma^2)$$

where *a* is set by `--weight_rpx` (default 1) and *b* by `--weight_ncontact`. Note that while the default value of `--weight_ncontact` in the standard score function is 0.001, a value of 5 is recommended for the `sasa_priority` score function. N is the number of contacting residues in the interface, scored based on a log normal distribution with a mean, $\mu$, set by `--weight_sasa` (default 1152 Å$^2$) and a tolerance level, $\sigma$, set by `--weight_error` (default 4). The resultant top-scoring configurations are biased towards the mean (**Fig 4A**), such that the buried SASA of the top docks at or very close to the `weight_sasa`, should such docks exist (**Fig 4B**). An artifact of this score function is that at higher target interface sizes, a set of high-scoring docks with small SASA estimate values emerge as a result of very small interfaces with high average *RPX* score; these outliers can be removed by the `filter_sasa` (see Additional Optional Filters below).

The `--weight_sasa` parameter may need to be modified depending on the docking problem. For example, cyclic docking might require a different `--weight_sasa` parameter than one- or two-component polyhedral group docking. The optimal `--weight_sasa` may be determined empirically for each architecture or docking problem by comparing independent docking trajectories and visually inspecting the results. Note that if the value is set to improbably high values (e.g., 99999), the search will fail rather than finding the largest interface, as docks near that SASA value do not exist. Note that this score function was fit using two-component polyhedral group architectures, so other architectures may need additional optimization of the variables. The development and optimization of this score function is described further in the S1 Text.

*Other score functions.* Additional variants of the standard score function are available, by replacing the *sum* of the maximum *RPX* scores at each residue pair considered in the *stnd* score function with the *mean* or *median* (**Table 2**). These two score functions partially remove the correlation between *ncontact* and total *RPX* score. Finally, two more functions were used in development of the `sasa_priority` score function that empirically estimated the relationship between *RPX* and *ncontact* with either a linear or exponential fit.

## Filtering docks

**Clustering.**   After docked configurations are scored, the results are clustered through redundancy filters. Redundancy checking is performed by the `filter_redundancy()` function, which performs a distance check on the transformed bodies (approximating an unaligned RMSD calculation) and discards similar transforms with distances below a user-

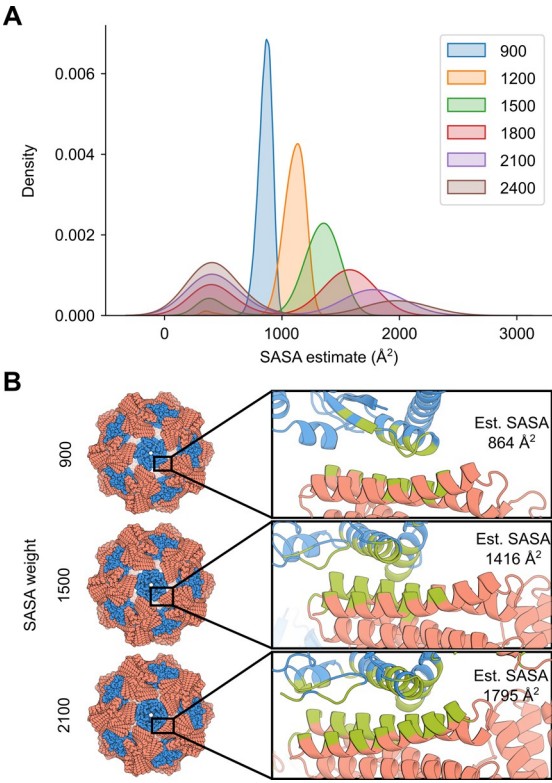

**Fig 4. Interface size bias by the sasa_priority score function. A.** 572 pairs of inputs were docked in a two-component icosahedral architecture at a--weight_sasa value of 900, 1200, 1500, 1800, 2100, and 2400, with total area under each curve normalized to 1. **B.** The interface of the top-scoring docked configuration for --weight_sasa value of 900, 1500, and 2100 is highlighted (green). Estimated buried SASA calculated using Rosetta for these docks are 864, 1416, and 1795 $Å^2$, respectively.

defined cutoff set by the `--max_bb_redundancy` option (default, 3 Å). Cluster size can be controlled by the `--max_cluster` option (default, no limit), which specifies the maximum number of clusters the docked transforms can be sorted into. As docks are sorted by score, only the highest-scoring dock from each cluster is kept. The redundancy filter returns an array of indices corresponding to the docked configurations that pass this filter.

**Additional optional filters.** We have developed a set of modular filters that can be applied post-docking to remove docks that do not meet certain requirements or to provide more information about the results. Currently available filters are:

**Table 2. List of additional score functions.**

| --function | Description |
|---|---|
| *stnd* | score = a * *RPX* + b * *ncontact*, where *RPX* is the sum of the max(motif score) across all residue pairs in a docked interface |
| *sasa_priority* | Function developed to bias interfaces to a certain size given user requirements. The `--weight_sasa` (default = 1152), `--weight_ncontact` (default = 0.01 but a value of 5 provides optimum scaling for this score function), and `--weight_error` (default = 4) flags must also be specified. |
| *mean* | Takes the mean of max(motif score), instead of sum() in the standard score function |
| *median* | Takes the median of max(motif score), instead of sum() in the standard score function |
| *exp* | scores = $RPX - 4.6679 * ncontact^{0.588}$ |
| *lin* | scores = $RPX - 0.7514 * ncontact$ |

- `filter_sscount:` Removes docks below a specified number of secondary structure (SS) elements in the docked interface

- `filter_sasa:` Removes docks outside a specified interface SASA using a similar method to the *sasa_priority* score function

New filters can be added without having to modify code in the *search* or *scoring* modules. At the time of publication, filtering is possible for architectures of the cyclic, dihedral, stacking, wallpaper, and polyhedral groups.

Filter behavior is controlled by a .yaml configuration file passed through the `--filter_config` option. This allows facile stacking of an arbitrary number of filters, including multiple instances of the same filter configured in different ways. Filters are defined with a key, or filter label, that can be any arbitrary string without whitespace. All filters have standard and filter-specific parameters. The standard parameters are a "type" parameter and a "confidence" parameter. The "type" parameter must exactly match the name of the filter in the RPXDock main code. The "confidence" parameter is a boolean that controls whether or not the filter will remove docks from the result. If confidence is *False*, the result will report values for all docks, including those that would have been removed had the confidence been set to *True*. Note that if confidence is *True*, a filter can potentially remove all of the results if none of them meet the thresholds, resulting in an empty result object. **S2** and **S3** **Tables** provides a list of all available filter-specific parameters.

## Result

After RPXDock has been executed, the result class outputs a zipped tarball .txz file and a .pickle file that stores i) the initial body object along with ii) the transforms and iii) associated score and filter values of each docked configuration in a concatenated xarray format. While the .pickle file is faster to access, it is Python version-dependent, so the .txz format is also returned as a version-agnostic output. Each of these output formats can be turned on or off using their respective options: `--save_results_as_tarball` and `--save_results_as_pickle`, which both default to *True*. With the the `dump_pdb()` function, the result object can output the resulting dock in the form of a .pdb file for any given model number, corresponding to the rank of the desired docked configuration by score. We have included an example Python script in the GitHub repository under `tools/dump_pdb_from_output.py` that demonstrates how to access score and filter information and regenerate docked configurations as .pdb files for any desired dock configuration from either file format. The `--overwrite_existing_results` option, which defaults to *False*, can be passed to overwrite existing outputs for file management purposes.

The top-scoring transforms can be directly output in .pdb format using the `--dump_pdbs` option. When used in combination with `--nout_top` N, which defaults to 10, the top-scoring N transforms can be output in .pdb format from the RPXDock result object. The user may be interested in saving disk space or for other reasons only saving the asymmetric unit (asu) of the resulting dock; this behavior can be set with the option `--output_asym_only`. The `--output_closest_subunits` option can be used in combination, which outputs a .pdb file containing the asu chains in positions that exhibit the highest motif contact count from the symmetric result (eg. the asu chains that are closest to each other in space) instead of the default asu chains in positions defined by each symmetry. This can be useful for visualization and for generating inputs for downstream steps in design pipelines.

## Results

We set out to experimentally evaluate symmetric one- and two-component structures with polyhedral group symmetry generated using RPXDock. Given a set of prevalidated homomeric scaffolds with cyclic symmetry, we generated docks using RPXDock, and the resulting interfaces were sequence-optimized via Rosetta sequence design [49]. Two one-component designs (T3-rpxdock-02, I3-rpxdock-71) and two two-component designs (O43-rpxdock-15, O43-rpxdock-HO11) with tetrahedral, octahedral, or icosahedral symmetry were examined by negative-stain electron microscopy and found to adopt the intended architecture (**Figs 5A–5D and S5A**). I3-rpxdock-71, while completely independently sampled and designed, resembles a dock previously sampled by RPXDock's predecessor, tcdock, indicating that the similar top results are identified by the new search algorithm [3]. We obtained a 3.7 Å resolution single-particle reconstruction of the two-component octahedral assembly O43-rpxdock-EK1 (PDB: 8FWD, EMD-29502) using cryogenic electron microscopy and found that it assembles to the intended structure with high accuracy (4 Å Cα root mean square deviation between all 48 chains of the original dock and cryoEM structure, **Figs 5E and S5B-S5F and S5 Table**). Together, these data confirm that docks generated using RPXDock can be designed to assemble in the intended configurations without disrupting the integrity of the starting scaffolds. Input .pdb files, docking and design scripts, and design models are provided in the tools/ directory available on the RPXDock GitHub page at https://github.com/willsheffler/rpxdock.

## Availability and Future Directions

### Setup and installation

At the time of publication, RPXDock has been verified to compile and function correctly on Linux-based operating systems. To set it up, a user must first clone the public repository of the full source code, which can be found at https://github.com/willsheffler/rpxdock, and set up a proper conda environment using the environment.yml file. Note that a user must obtain a pyrosetta license (free for non-profit users) and update the username and password fields for their pyrosetta license in the environment.yml file before creating the environment. Users may need to also install additional packages in their conda environment such as pyyaml to properly build the application. To build and compile the codebase with the newly created conda environment, a user may simply run the pytest command using a gcc>9-compatible compiler.

To verify that the code compiled properly, execute rpxdock/app/dock.py --help in the new conda environment. The output should provide a list of options that are relevant for docking (S1 Table). Note that several options are still experimental in nature and therefore are not described fully in this publication. For a template of how to set up a simple RPXDock run, please refer to the available example provided in tools/dock.sh in the GitHub repository.

## Discussion

RPXdock provides a powerful and general route to modeling, sampling, and scoring symmetric protein complexes across multiple symmetric architectures. Docking monomeric and oligomeric building blocks into higher-order symmetric complexes followed by protein-protein interface design is an established and successful paradigm for accurately creating novel self-assembling protein nanomaterials [2,4,17,19–22]. While deep learning-based generative models have recently proven successful in designing de novo oligomers [50] and small nanocages [51], the ability of RPXdock to use experimentally verified or previously designed scaffolds in a stepwise manner enables the use of specific building blocks that have optimal features for specific applications [1,39,40,43,44]. The RPXdock code can accommodate specific user

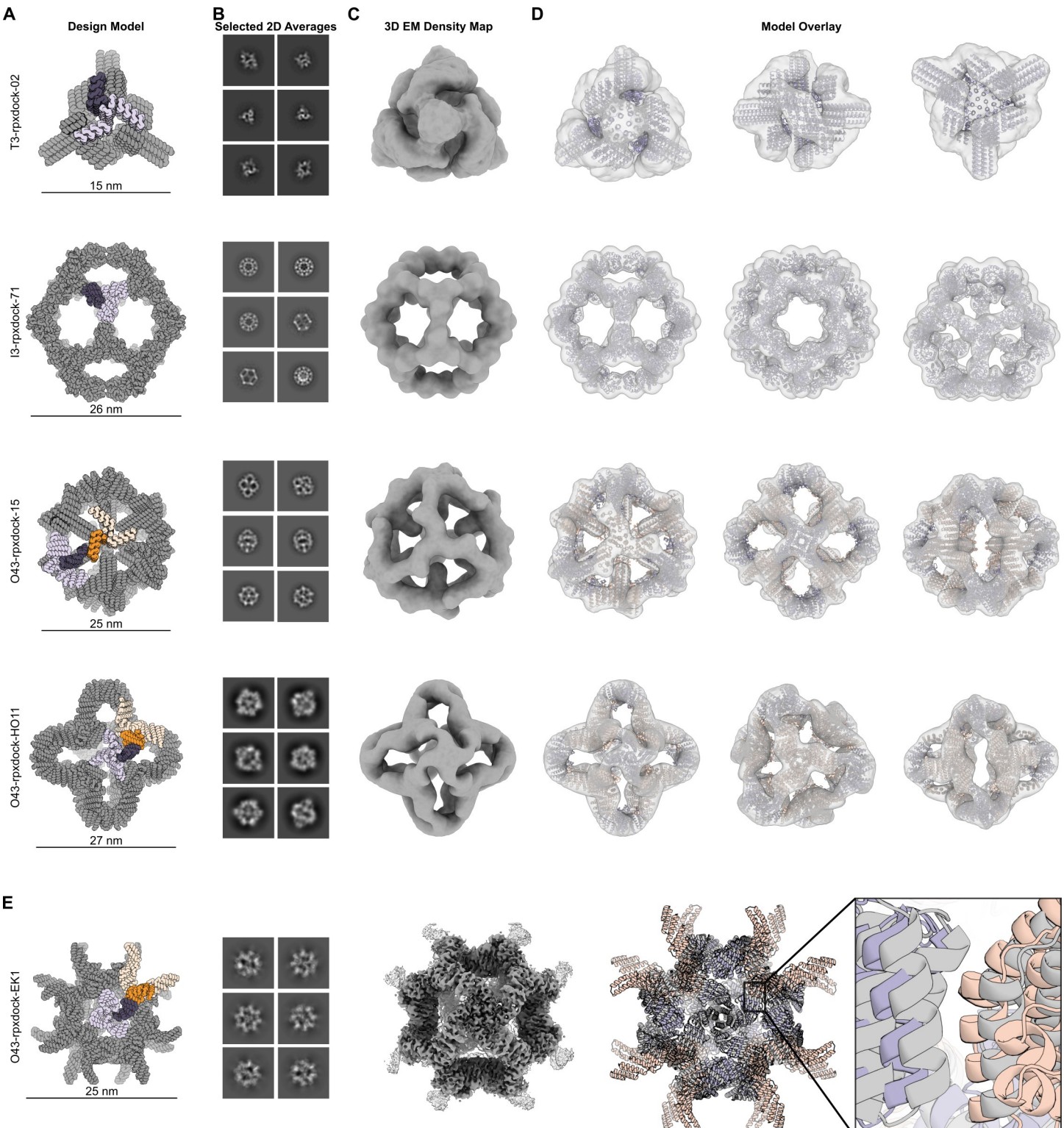

**Fig 5. Docking and characterization of one- and two-component polyhedral assemblies using RPXDock. A**. Models of one- and two-component docked polyhedral assemblies with the oligomeric building blocks in purple and orange. The asymmetric unit of each assembly, comprising one subunit of each building block, is colored dark purple and dark orange. **B.** Reference-free 2D class averages from negative stain electron microscopy. Each assembly is viewed along several axes of symmetry. **C.** 3D density maps reconstructed from selected 2D class averages. **D.** Overlays of each design model fit into its 3D density map, confirming that each design assembles to the architecture identified by RPXDock. **E.** Characterization of the two-component octahedral assembly O43-rpxdock-EK1 by cryogenic electron microscopy. The design

model is colored as in A). To the right are representative 2D class averages showing different axes of symmetry and a reconstructed 3D map at 3.7 Å resolution. The overlay of the original dock (orange and purple) with the model built from the 3D reconstruction (gray) shows 4 Å Cα root mean square deviation between the original dock and cryoEM structure over 48 chains.

requirements for complex docking problems, and the efficiency at which high-quality docks are found has been greatly improved compared to its predecessors (tcdock; sicdock; sicaxel [17,19,39,40]) due to the hierarchical search and scoring algorithms. While new capabilities are continuously under development, the core software structure is complete and robust, and has already been successfully applied to a number of symmetric docking and design problems in addition to the structures presented here ([29–31]. Any future modifications and new modules added to the RPXDock application will be updated via the GitHub repository: https://github.com/willsheffler/rpxdock.

## Supporting information

**S1 Fig. Hierarchical docking performance test. A**. A 2-dimensional illustration of a hierarchical search grid with samples searched at the highest resolution in blue vs. a complete search grid at the same resolution. In this test dataset, ~$10^8$ total docks were sampled. **B**. A cumulative distribution of the fraction of the total search space that needs to be sampled in order to recover the top 1, 10, 100, and 1000 docks from this dataset.
(TIF)

**S2 Fig. NContact and SASA are highly correlated.** As such, we parameterized an ncontact score term with respect to computationally measured interface size, SASA.
(TIF)

**S3 Fig. Parameterization of an ncontact score term as a function of interface size, SASA, results in a log-normal distribution with a maximum ncontact score term at a user-input SASA regardless of standard deviation.**
(TIF)

**S4 Fig. Empirical derivation of the ncontact score term weight. A.** Score as a function of ncontact across various ncontact weights. **B.** Mean *RPX* as a function of ncontact. **C.** Mean *RPX* as a function of ncontact weighting plotted for interface sizes from Number of unique contacts = 5–55. **D.** Total number of passing designs out of 960 docks for each weighting and fraction. **E-F.** Computational design metrics as a function of ncontact weight for top-, middle-, and bottom-ranked designs for **E.** ddG, and **F.** SASA. **G.** The top dock with I32 icosahedral symmetry for, left to right, ncontact weight 1, 3, 5, 7, 9.
(TIF)

**S5 Fig. nsEM and CryoEM data and associated plots of one- and two-component polyhedral self-assembling proteins from RPXDock. A.** Representative raw nsEM micrographs of one- and two-component polyhedral self-assembling proteins from RPXDock. Scale bar = 100 nm **B.** Representative raw CryoEM micrograph showing good particle distribution and contrast of (Scale Bar = 100 nm). **C.** CryoEM local resolution map of O43-rpxdock-EK1, with the sharpened map at two different contour levels, using a tight mask, and calculated using an FSC value of 0.143. **D.** Local resolution estimates of the unsharpened map, also at two different contour levels (FSC = 0.143). The protruding arms of the designed cage only start to become visible at very low contour levels. Local resolution estimates range from ~3.2 Å at the core to >4.0 Å along the periphery of the extended arms due to a high degree of flexibility within this region. **E.** Global resolution estimation plot. **F.** Orientational distribution plot demonstrating

near-complete angular sampling.
(TIF)

**S1 Table. All RPXDock command line options.**
(XLSX)

**S2 Table. SASA estimate filter parameters.**
(XLSX)

**S3 Table. SScount filter parameters.**
(XLSX)

**S4 Table. Design construct renaming and input pdb files.**
(XLSX)

**S5 Table. CryoEM data collection and refinement statistics.**
(XLSX)

**S1 Note. Protein Sequences of validated RPXDock designs from Fig 5.**
(DOCX)

**S1 Text. Supplemental Information.**
(DOCX)

## Acknowledgments

We thank Lance Stewart, Christian Richardson, Derrick Hicks, Stacey R. Gerben, Ryan Kibler, George Ueda, and Jorge Fallas for helpful discussions, Shingo Honda for providing feedback on the manuscript, and Luki Goldschmidt and Patrick Vecchiato for computational resource management.

## Author Contributions

**Conceptualization:** William Sheffler, Erin C. Yang, Quinton Dowling, Yang Hsia, Alena Khmelinskaia, Neil P. King, David Baker.

**Data curation:** William Sheffler, Erin C. Yang, Quinton Dowling, Yang Hsia, Chelsea N. Fries, Jenna Stanislaw, Mark D. Langowski, Marisa Brandys, Zhe Li, Rebecca Skotheim, Andrew J. Borst, Alena Khmelinskaia.

**Formal analysis:** William Sheffler, Erin C. Yang, Quinton Dowling, Yang Hsia, Chelsea N. Fries, Rebecca Skotheim, Andrew J. Borst, Alena Khmelinskaia.

**Funding acquisition:** Neil P. King, David Baker.

**Software:** William Sheffler, Erin C. Yang, Quinton Dowling, Yang Hsia, Chelsea N. Fries, Jenna Stanislaw, Mark D. Langowski, Alena Khmelinskaia.

**Supervision:** Neil P. King, David Baker.

**Writing – original draft:** William Sheffler, Erin C. Yang, Quinton Dowling, Yang Hsia.

**Writing – review & editing:** William Sheffler, Erin C. Yang, Yang Hsia, Chelsea N. Fries, Alena Khmelinskaia, Neil P. King, David Baker.

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
