## [Decision Letter · Decision Letter 0]

15 Dec 2022

Dear Dr. Baker,

Thank you very much for submitting your manuscript "Fast and versatile sequence-independent protein docking for nanomaterials design using RPXDock" for consideration at PLOS Computational Biology.

As with all papers reviewed by the journal, your manuscript was reviewed by members of the editorial board and by several independent reviewers. In light of the reviews (below this email), we would like to invite the resubmission of a significantly-revised version that takes into account the reviewers' comments.

We are in agreement with the reviewers that this is important work. We are also in agreement with the reviewer that the manuscript would be much clearer with some example of an application. Though we understand that many applications would be publishable in their own right, we hope there is something that could illustrate RPXDock usage.

We cannot make any decision about publication until we have seen the revised manuscript and your response to the reviewers' comments. Your revised manuscript is also likely to be sent to reviewers for further evaluation.

Sincerely,

Joanna Slusky, Ph.D.

Academic Editor

PLOS Computational Biology

Lucy Houghton

Staff

PLOS Computational Biology

I am in agreement with the reviewers that this is important work. I am also in agreement with the reviewer that the manuscript would be much clearer with some example of an application. Though I understand that many applications would be publishable in their own right, I hope there is something that could illustrate RPXDock usage.

Reviewer's Responses to Questions

**Comments to the Authors:**

Reviewer #1: The paper describes a new computational procedure for docking proteins into user-defined oligomeric symmetries. A number of techniques addressing this problem have been developed by others. However, the procedure described in the paper emphasizes flexibility of its application to a variety of tasks, and utility for the practical use in protein design, including completeness of user's documentation, accessibility to the users community, etc. The procedure is properly benchmarked and described in great detail. As such it should be a useful addition to the toolchest of protein design community.

A small technical glitch - Supplemental Figure S1 is placed in the main text, instead of the Supplement.

Reviewer #2: This paper is the description of RPXDock, a software package for sequence-independent rigid-body protein docking across a wide range of symmetric architectures, suitable for large constructs of 1 or 2 types of proteins towards desired configurations. The software is novel and potentially very useful.

However, I consider it a problem that while the description of the methods and use options is complete and very detailed, there is not a single example demonstrating the use of the software. I think this is a missing component. The paper states that “RPXDock was used to successfully design cyclic oligomers (Gerben et al., submitted), one-component nanocages (Wang et al. 2022), two-component nanocages (Li et al., submitted; Huddy et al., in preparation; Dosey et al., in preparation), and even larger pseudo-symmetric nanomaterials (Dowling et al., in preparation; Lee et al., in preparation), establishing its utility and generality.” Thus, the only paper currently available is Wang et al. “Improving the Secretion of Designed Protein Assemblies through Negative Design of Cryptic Transmembrane Domains.” in preprint, and it focuses on the analysis of the designed assemblies rather than on their design. All the other paper are submitted or in preparation, and do not demonstrate the methodology. Thus, the performance of the method can be fully understood only by downloading and installing the software, which makes it difficult to form an informed opinion. I understand that the authors may want to publish the methodology prior to application papers, but I think including some demonstration of results would make this paper more interesting to readers.

**Have the authors made all data and (if applicable) computational code underlying the findings in their manuscript fully available?**

Reviewer #1: Yes

Reviewer #2: Yes

PLOS authors have the option to publish the peer review history of their article (what does this mean?). If published, this will include your full peer review and any attached files.

Reviewer #1: No

Reviewer #2: No
---

## [Editor Report · Decision Letter 1]

11 Feb 2023

Dear Dr. Baker,

Thank you very much for submitting your manuscript "Fast and versatile sequence-independent protein docking for nanomaterials design using RPXDock" for consideration at PLOS Computational Biology.

Your manuscript was reviewed by members of the editorial board.  We would like to invite the resubmission of a significantly-revised version that takes into account the following concerns:

The experimental validation provided is extremely impressive and illustrate well the functionality of the algorithm. However, more detail needs to be provided for how the experimental validation was preformed  for usability/reproducibility.  Please include details of:

1) what proteins were used 

2) which parts of the protein were redesigned using Rosetta or ProteinMPNN

3) which of those two methods was used for each protein

and 4) the parameters that were used in each case. 

We cannot make any decision about publication until we have seen the revised manuscript and your response to the reviewers' comments. Your revised manuscript is also likely to be sent to reviewers for further evaluation.

Sincerely,

Joanna Slusky, Ph.D.

Academic Editor

PLOS Computational Biology

Nir Ben-Tal

Section Editor

PLOS Computational Biology
---

## [Editor Report · Decision Letter 2]

9 Apr 2023

Dear Dr. Baker,

We are pleased to inform you that your manuscript 'Fast and versatile sequence-independent protein docking for nanomaterials design using RPXDock' has been provisionally accepted for publication in PLOS Computational Biology.

Best regards,

Joanna Slusky, Ph.D.

Academic Editor

PLOS Computational Biology

Nir Ben-Tal

Section Editor

PLOS Computational Biology

---

## [Editor Report · Acceptance letter]

17 May 2023

PCOMPBIOL-D-22-01558R2 

Fast and versatile sequence-independent protein docking for nanomaterials design using RPXDock

Dear Dr Baker,

I am pleased to inform you that your manuscript has been formally accepted for publication in PLOS Computational Biology. Your manuscript is now with our production department and you will be notified of the publication date in due course.

With kind regards,

Anita Estes
